# Synoptic perspectives on pollutant transport patterns observed by satellites over East Asia: Case studies with a conceptual model

Hyun Cheol Kim<sup>1,2</sup>, Soontae Kim<sup>3</sup>, Seok-Woo Son<sup>4</sup>, Pius Lee<sup>1</sup>, Chun-Sil Jin<sup>5</sup>, Eunhye Kim<sup>3</sup>, Byeong-Uk 5 Kim<sup>6</sup>, Fong Ngan<sup>1,2</sup>, Changhan Bae<sup>3</sup>, Chang-Keun Song<sup>7</sup>, and Ariel Stein<sup>1</sup>

1 Air Resources Laboratory, National Oceanic and Atmospheric Administration, College Park, MD

2 Cooperative Institute for Climate and Satellites, University of Maryland, College Park, MD

3 Department of Environmental and Safety Engineering, Ajou University, Suwon, Korea

4 Seoul National University, School of Earth and Environmental Sciences, Seoul, Korea

5 National Centers for Environmental Prediction, National Oceanic and Atmospheric Administration, College Park, MD 6 Georgia Environmental Protection Division, Atlanta, GA 7 National Institute of Environmental Research, Incheon, Korea

Correspondence to: Hyun Cheol Kim (hyun.kim@noaa.gov)

- Abstract. We demonstrate that daily pollutant transport patterns in East Asia are visible from satellite images when inspected with corresponding synoptic weather analyses. Transport pathways of air pollutants in East Asia are investigated using satellite observations, surface weather charts, and chemical-transport model simulations. It is found that during cool season (fall to spring), pollutant transports in East Asia are largely determined by synoptic weather patterns associated with high pressure system over southern China, which is extended from the Siberia High, and low pressure system over
- Manchuria, which is initiated by Altai-Sayan cyclogenesis. Based on the relative location and strength of these weather systems, three types of synoptic weather patterns that may contribute to pollutants transport in East Asia, especially in China and Korea, are identified: i.e., (1) a strengthening of the Siberian High and its southeastward propagation; (2) a high-pressure system over southern China followed by a frontal passage associated with a northern low-pressure system; and (3) a stagnant high-pressure system over southern China. For all three patterns, the high-pressure system in southern China is essential for
- the development of regional air pollution, while frontal activities associated with low-pressure system provide a forcing mechanism to transport those pollutants eastward or southeastward. Observed and simulated surface PM distributions show good agreement in both aerosol optical depth and NO2 column density further implying that anthropogenic emissions also contribute to regional events of high surface PM concentrations. It is argued that the quasi-periodic migration of synoptic weather systems in East Asia works as an efficient pump of pollutants; i.e., regional air pollutions developed under high-
- pressure systems are transported downstream by low-pressure systems.

# 1 Introduction

Regional air quality is a rising concern in East Asia (e.g. China, Korea, and Japan), where the chemical environment has rapidly changed due to fast economic growth and industrialization, especially in China (Guan et al., 2009; Park and Kim, 2014; Richter et al., 2005; Zhang et al., 2007). The rapid and varied industrial development, the use of coal for power generation, and the increasing number of vehicles on the roads have increased the emission of pollutants in the region, especially particulate matter (PM) and its precursors. In January 2013, the highest instantaneous concentration of PM<sub>2.5</sub> reached over 1000 μg/m<sup>3</sup> in heavily polluted areas of Beijing (Zhang et al., 2014). Woes continue as of 2015, with levels of daily PM<sub>2.5</sub> concentration at the US embassy in Beijing recorded as high as 450 μg/m<sup>3</sup> (February 25, 2014) and 537 μg/m<sup>3</sup> (December 25, 2015) (http://www.stateair.net/web/post/1/1.html), almost 20 times higher than the standard recommended by

10 the World Health Organization (WHO, 2005).

Korea also has similar challenging and ongoing regional air-quality issues, especially in the Seoul Metropolitan Area (SMA), the most populated and industrialized region in Korea. The Korean government legislated a special Act in 2005 implementing a series of strict policies to reduce air-pollutant emission, with the goal to attain  $PM_{10}$  and nitrogen dioxide (NO<sub>2</sub>) concentration levels of 40 µg/m<sup>3</sup> and 22 ppbv, respectively, by 2014, compared to 69 µg/m<sup>3</sup> and 38 ppbv, respectively,

- in 2003 (UN, 2006). Continuous efforts from the Korean government and society have indeed improved the annual mean  $PM_{10}$  (WMO, 2012); however, one yet-unsolved issue is that the SMA still experiences frequent episodic regional pollution events and continuous degradation of visibility, especially during the winter (Oh et al., 2015). Recent studies have suggested that the long-range transport of emissions or pollutants from neighboring countries—especially from China— has partly contributed to the SMA's air pollution problem (Jeong et al., 2011; Lee et al., 2008; Lee et al., 2014; Moreno et al., 2012;
- Ou-Yang et al., 2013; Verstraeten et al., 2015). The mechanism of these transport pathways, however, has not been fully understood yet.

In this study, we revisit the relationship between synoptic weather patterns and the development and transport of air pollutants in East Asia (Ding et al., 2009; Lee et al., 2011b, 2013; Oh et al., 2015; Zhang et al., 2016). Only the cool season (fall to spring) is considered as air pollutions are severe in this season while pollutants are quickly washed out during Asian

summer monsoon period. Unlike in previous studies, all available satellite datasets, synoptic weather maps and chemistry transport model output in the region are used. In order to directly compare datasets with varying resolutions, all data are carefully re-projected on the model grids using newly developed geospatial data processing technique. We begin with a conceptual model that is based on the cool season synoptic weather systems in East Asia. Section 2 presents

three dominant synoptic weather patterns, potentially associated with pollutant transport in East Asia, that are classified by

30 considering the location and strength of high-pressure and low-pressure systems in the surface weather maps. In Section 3, short descriptions of satellite and model data used in this study are provided. In Section 4, we illustrate the three synoptic weather patterns that may control regional-scale pollutant transport patterns, presenting satellite observations and model

20

simulations. The possible contribution of Chinese anthropogenic emissions to surface PM concentrations in Korea is also briefly discussed. Finally, summary and conclusions are presented in Section 5.

# 2 Classification of pollutant transport patterns in East Asia

- The synoptic weather patterns have been extensively examined to identify transport pathways of pollutants and/or of their precursors, including Asian Dust, in East Asia (Chun et al, 2001; Chung, 1992; Husar et al., 2001; Iwasaka et al., 1983; Kim et al., 2005; Kim, 2008; Merrill et al., 2004; Sun et al., 2013; Tan et al., 2012). Among others, Kim et al. (2001a, 2001b) studied the transport and evolution of wintertime Asian Dust observed in Korea, and the transport of SO<sub>2</sub> and aerosol over the Yellow Sea. Kim et al. (2005) used Hybrid Single Particle Lagrangian Integrated Trajectory (HYSPLIT, Stein et al., 2015) back-trajectories and performed a cluster analysis during the ACE-Asia campaign.
- The pollutant transport patterns are important not only for regional air quality but also for global climate variability and change. Wang et al. (2014), who used a multiscale, global aerosol/climate model to assess the effects of Asian pollution outflows on regional climate and global atmospheric circulation, demonstrated that long-range transport of Asian pollutants and the resulting variations in aerosol optical depth, cloud-droplet concentration, and cloud- and ice-water paths can enhance cloud radiative forcings and increase precipitation and poleward heat transport. This clearly suggests that pollutant transport
- in East Asia is not simply a regional air quality issue. However, the processes that leads to the transport, transformation, and removal of pollutants in East Asia still remain uncertain (Dickerson et al., 2007).

# 2.1 Synoptic weather systems in East Asia

During the Northern Hemisphere cool season, synoptic weather systems in East Asia are mainly characterized by the persistent Siberian high and the transient development of low-pressure systems on the lee side of mountains. A simplified diagram, highlighting migratory synoptic weather systems in northeast Asia, is shown in **Figure 1**.

The Siberian High, which is primarily formed and maintained by intense surface cooling, is quite persistent during the cool season (Cohen et al., 2001; Gong & Ho, 2002; Wu & Wang, 2002). However, it also exhibits significant intra-seasonal variability. An episodic expansion of the Siberian High deep into southern China often causes a cold surge in East Asia (Compo et al., 1999; Garreaud, 2001; Park et al., 2013). Such an expansion of continental high-pressure system, along with

25 migrating low-pressure systems in northern China has been considered to be a dominant pathway driving Asian pollutions off the continent. The associated cold advection also helps gas-to-particle heterogeneous processes to produce more secondary products in terms of PM concentrations (Behera and Sharma, 2010).

A prevailing high-pressure system, or near-surface anticyclone, is known to play an important role in the development of local air pollution. In general, high-pressure system accompanies a clear sky, calm wind, and subsiding air, lowering the

30 chances of pollutants being washed out by precipitation or blown away by strong winds (Chen et al., 2008; Cheng et al., 2007; Frioud et al., 2003; Im et al., 2008; Tai et al., 2010; Wei et al., 2011). The relative location of high-pressure system

with respect to the source regions of air pollutants is also important (Wong et al., 2007). When the high-pressure system resides near the southeastern China or the Yellow Sea, that is south of the heavily polluted areas (e.g., the Beijing-Tianjin-Hebei region), the anticyclonic outflow on the northern side of the high-pressure system, that is westerly or northwesterly, creates the perfect conditions for transporting highly polluted air to Korea and Japan.

- Regional air pollution is also influenced by the travelling low-pressure systems or cyclones. In East Asia, Altai-Sayan cyclones, which are initiated over the Mongolian Plateau on the lee side of the Altai-Sayan Mountains (Chen et al., 1991), tend to travel southeast or east-southeast towards northeast China or Manchuria. These transient weather systems are known as one of the major processes initiating Asian Dust, since their arid characteristics may trigger wind erosion across a wide area and the associated cold frontal activities may lift dust from the surface (Chung, 1992). Likewise, Liu et al. (2003), who
- examined carbon monoxide pathways from chemistry transport model simulations and aircraft measurements from the Transport and Chemical Evolution over the Pacific (TRACE-P) mission, also demonstrated that the major process driving Asian pollutions outflow in the spring is frontal lifting at the leading edge of southeastward-moving cold front and transport in the boundary layer behind the cold front. Ding et al. (2015) also demonstrated uplifting of CO from biomass burning and anthropogenic sources to the free troposphere in East Asia, using satellite, trajectory dispersion model and global chemistry
- transport model.

However, unlike high-pressure system, an intensified low-pressure system plays a complicated role in the regional air quality because it also operates as a scavenging and cleaning process for most primary and secondary anthropogenic pollutants (e.g., Wei et al., 2011b). Therefore, determining the relative location and timing of these systems is crucial to identify the pollutant transport patterns and their impact on regional air quality. In the next section, we conceptually classify the pollutant transport

20 patterns in East Asia, by considering the relative location and intensity of high- and low-pressure systems.

# 2.2 Conceptual classification

In this section, we classify the pollutant transport patterns in East Asia, focusing on China and Korea. We first start with a conceptual understanding of regional flow patterns and emission source locations. This approach focuses on more conceptual and empirical understanding (based on daily forecast experiences), rather than numerical classification. As shown later, inspection of daily variations of Aerosol Optical Depth (AOD) and NO<sub>2</sub> column density from satellite images, hourly variations of AOD, NO<sub>2</sub> vertical column density, and PM distributions from chemistry transport model, and surface weather map analyses reveals that the strength and location of anticyclonic systems over southern China (usually expanded from the Siberian High) and those of cyclonic systems over Manchuria (initiated by Altai-Sayan cyclogenesis) are the two major components that determine regional pollutant transport patterns. Objective clustering analyses in the previous studies (Kim et

30 al., 2005; Kim et al., 2014; Lee et al., 2011; Park et al., 2013) are also carefully considered. Kim et al. (2005) used cluster analysis of HYSPLIT backward trajectories arriving at the Gosan site in Jeju island, Korea, to classify five pollutant transport patterns. Their study suggested that except for Cluster 1, which originates from the Pacific region, 78.3% of trajectories come through westerly (Cluster 2, 6.7%) or northerly (Clusters 3, 4, and 5, together 71.7%). Lee et al. (2011)

used FLEXTRA trajectory model to classify 254 trajectories during high PM concentration events in Korea into the three types: i.e., long-range transport type (28%), in-between type (41.3%), and local type (30.7%). These types are compatible with the classification types used in this study.

Conceptual classifications of synoptic weather patterns, that are likely associated with pollutant transport in China and

- Korea, are shown in Figure 2. Based on sea level pressure (SLP), the following three types are identified: i.e., (1) a strengthening of the Siberian High and its southeastward expansion or propagation (Figure 2a); (2) a high-pressure system over southern China, followed by a cold frontal passage associated with a low-pressure system over northern China (Figure 2b); and (3) a stagnant high-pressure system over southern China or near the Yellow Sea (Figure 2c). In all cases, westerly or northwesterly flow near the surface is crucial for the development of pollutant events in eastern China and Korea (Chen et al. 2b).
- al., 2008; Lee et al., 2013). In (2), Manchurian low-pressure systems (e.g., those formed by Altai-Sayan cyclogenesis) provide a forcing mechanism for pollutant transport and/or cleaning at the dissipation stage of high-pollutant episodes. In East Asia, travelling synoptic weather systems are very common in cool season because of baroclinic instability (Park et al., 2013). When a high-pressure system prevails, it provides favorable meteorological conditions for the development of high-pollution events. On the other hand, low-pressure systems, especially those with strong cold frontal activity, push
- continental (e.g., Chinese) pollutants out to neighboring countries or to the western Pacific. Each synoptic weather pattern is discussed in more detail in Section 4 with observational evidences.

#### 3 Data and Methodology

Both chemical transport model outputs and observational datasets are analyzed for the period of November 2012 to May 2014. As noted earlier, only the two cool seasons, November to May, are considered.

### 20 3.1 Synoptic weather chart and surface PM<sub>10</sub> data

Surface  $PM_{10}$  concentrations and meteorological observations (e.g., wind speed and surface pressure) were obtained from the NIER and KMA, respectively. Three-hourly surface weather charts were also obtained from KMA. Close investigation of surface weather charts and satellite datasets is essential to understand the role played by synoptic weather systems in the development and transport of surface air pollutions. This kind of analysis is not very common in the literature because

operational surface weather maps are typically provided as graphic files. In this study, we utilized a graphical technique to blend satellite images or model output with KMA weather charts. This composite technique, which is based on a Geographic Information System (GIS) georeferencing technique, is useful when only graphical analyses, taken from different institutions, are available without digitized data sets. For a technical note about this technique, refer to the Appendix.

## 3.2 Satellite data

Various satellite and in-situ observations are analyzed in the present study. They include Moderate Resolution Imaging Spectroradiometer (MODIS) Aerosol Optical Depth (AOD), Global Ozone Monitoring Experiment (GOME)-2 and Ozone Monitoring Instrument (OMI) tropospheric NO<sub>2</sub> vertical column density (VCD), surface PM<sub>10</sub> concentrations obtained from

5 the National Institute of Environmental Research (NIER), and weather maps from Korea Meteorological Administration (KMA). Key aspects of each datasets are briefly described below.

**MODIS AOD:** The MODIS aerosol product globally monitors ambient aerosol optical properties, such as optical thickness and aerosol size distribution, over the oceans and continents (http://modis-atmos.gsfc.nasa.gov/MOD04\_L2/index.html). Daily level 2 products are archived and utilized to generate AOD for model grid cells. We used both Terra and Aqua

products (MOD04 & MYD04 collection 6; Levy et al., 2013), which overpass at 10:30 am and 1:30 pm local times, respectively, with 10 km x 10 km horizontal resolution.

**NO2 VCD:** The anthropogenic emissions fluxes are estimated by using GOME-2 and OMI tropospheric NO<sub>2</sub> VCD, retrieved by the Royal Netherlands Meteorological Institute (KNMI). The GOME-2 sensor, which is onboard the EUMETSAT MetOp-A and MetOp-B satellites, conducts nadir measurements around 9:30 am local time with footprints of

- 40 × 80 km<sup>2</sup> (or, since 2013, 40×40 km<sup>2</sup> for GOME-2A). By contrast, OMI, onboard the NASA's Earth Observing System Aura satellite, has 1:30 pm local overpass time with 13 × 24 km<sup>2</sup> pixel resolution. Data were downloaded from the European Space Agency's (ESA) Tropospheric Emission Monitoring Internet Service (TEMIS; <u>http://www.temis.nl/airpollution/no2.html</u>). TM4NO2A version 2.3 data were used for GOME-2, and DOMINO version 2.0 data were used for OMI. We disregarded contaminated data pixels or those with cloud fractions over 40% using quality
- flags. The Differential Optical Absorption Spectroscopy (DOAS) technique was used for both products. Details regarding the NO<sub>2</sub> column-retrieval algorithms and error analysis were described in Boersma et al. (2004, 2007). Pixels are converted to CMAQ domain grids using a conservative re-gridding method (Kim et al., 2016b).

The use of NO<sub>2</sub> VCD data is noteworthy. One might question why we used NO<sub>2</sub> VCD for the pollutant transport study, since NO<sub>2</sub> is rarely used in chemical-transport studies because of a very short lifetime. This short life time, however, allows us to interpret NO<sub>2</sub> VCD as fresh anthropogenic emissions.

#### 3.3 Chemical transport model

To estimate the generation and transport of pollutants, a chemistry transport model is integrated. Specifically, gas and aerosol concentrations over East Asia are simulated by using a Weather Research and Forecasting Model (WRF)–Sparse Matrix Operator Kernel Emission (SMOKE)–Community Multiscale Air Quality (CMAQ) modeling system. We used WRF

version 3.3.1 (Skamarock and Klemp, 2008) for the meteorology simulation, initiated with the NCEP GFS 0.5° × 0.5° global product over a 27-km East Asia domain. Terrain and surface land types are taken from 90-m Shuttle Radar Topography Mission (SRTM) Digital Elevation Model (DEM) and Korean Ministry of Environment Land Use/Land Cover data.

5

Chemical transport simulations are performed with CMAQ version 4.7.1 (Byun and Schere, 2006) with the AERO5 aerosol module and Statewide Air Pollution Research Center version 99 (SAPRC99; Carter, 1999). See Table 1 for the summary of the model configuration. As a preprocessor for the CMAQ simulation, the Meteorology-Chemistry Interface Processor (MCIP) version 3.6 is used. The 2006 Intercontinental Chemical Transport Experiment-Phase B (INTEX-B 2006) emissions inventory (Zhang et al., 2009) was used, except inside South Korea, where we instead used the Clean Air Policy Support System (CAPSS) 2007 emissions inventory (Lee et al., 2011). The Model of Emissions of Gases and Aerosols from Nature (MEGAN; Guenther et al., 2006) was used to prepare biogenic emissions. Detailed descriptions on the model configurations

#### 4 Observed and modeled pollutants transport patterns in East Asia

and model performance evaluations are available at Kim et al., (2016a).

#### 10 4.1 Type 1: Expansion of the Siberian High

Type 1 pattern is characterized by a prevailing high-pressure system in southern China (Fig. 2a) which affects both the development of local air pollutions and their subsequent transport. As a high-pressure system locates over southern China or the Yellow Sea, it provides an anticyclonic outflow that causes strong eastward transport of pollutants from the Chinese east coast (e.g., Shanghai) or northern China (e.g., Beijing) to Korea and Japan. This pattern is often accompanied by weak low-

15 pressure systems in the Manchuria. A cyclonic flow in the southern part of a low-pressure system can reinforce eastward transport of air pollutants.

When the high-pressure system remains attached to the Siberian High, its long-range transport pattern is more southeastward to eastward, passing through Mongolia as a possible pathway of wintertime Asian Dust transport to Korea (Kim and Park, 2001; Kim et al., 2001). Note that Type 1 is compatible with Cluster 2 in Kim et al. (2005) and the long-range transport type

- in Lee et al. (2011). **Figure 3** shows satellite images and modeled pollutants for a case study of Type 1. A prominent highpressure system over southern China and a weak low-pressure system in northern China cause a strong eastward pollutant transport from China to Korea during December 30–31, 2013. As the high-pressure system is expanded from the Siberian High, the northwesterlies from Mongolia to the Chinese east coast become dominant, resulting in strong outflow from the continent. These flow patterns are well represented in the CMAQ PM<sub>10</sub> simulations as evident from the enhanced PM<sub>10</sub>
- concentrations over the Yellow and East China Seas. MODIS AOD shows a similar spatial distribution to CMAQ  $PM_{10}$  concentrations. GOME-2 and OMI NO<sub>2</sub> column density further show a good agreement over the east coast and the Yellow Sea, although they disagree in the southern part of the region. During this period, Korea experienced high surface  $PM_{10}$  concentrations, with a report of an Asian Dust (see Fig. 9 as discussed later). As such, the December 30–31, 2013 high  $PM_{10}$  concentration episode in Korea is likely caused by the combined impact of Chinese anthropogenic emissions and Asian Dust
- transport.

Figure 4 depicts another example of Type 1 pollutant transport pattern. In this case, an expanded high-pressure system is detached from the Siberian High, showing more anticyclonic flow patterns over the Yellow Sea. Again, satellite-observed

 $NO_2$  column densities, especially those from GOME-2, show an excellent agreement with the simulated distribution of surface  $PM_{10}$  concentrations, suggesting that anthropogenic emissions have likely contributed to high  $PM_{10}$  concentrations. Animation of this figure is available in the supplementary materials.

# 4.2 Type 2: Cold frontal passage

- We separated Type 2 from Type 1 because Type 2 involves two sequential stages: i.e., (1) a prominent high-pressure system over southern China and (2) the growth of a dominant low-pressure system over northern China (Fig. 2b). Type 2 also differs from Type 1 in terms of pollutant transport direction. While Type 1 shows dominant westerly (and northwesterly inland from the Siberian High), Type 2 is mostly associated with northerly (and slightly northwesterly), related to the cold frontal passage of the low-pressure system. It is also highlighted by a radially shaped (or narrow-banded) pollutant transport that is
- pushed by the passing cold front. Another important aspect of Type 2 pattern is its lifting capability, which is also associated with strong frontal activity.

**Figure 5** shows the typical development of Type 2 synoptic weather pattern and pollutants during November 4–7, 2013. On November 4, a dominant high-pressure system over inland China provided a favorable condition for the consolidation of pollutants. There is also pollutant signal near or north of Beijing. On November 5, the high-pressure system was still strong,

- but its center was displaced to the Yellow Sea. With help from the low-pressure system around Mongolia, the flow pattern across Beijing becomes southwesterly, pushing pollutants near Beijing northwestward and forming a heavily polluted air band up to Harbin or to Russian territory. On the next day, the low-pressure system strengthened, beginning to push condensed pollutants southward through the northerly winds associated with the cold front. Lastly, on November 7, the lowpressure system travelled eastward, and northerly winds behind the cold front pushed pollutants from the Manchurian region
- down to the southern boundary of Korea, resulting in a long and narrow band of pollutants from Shanghai to the Korean strait, and to the northern Hokkaido, Japan. This narrow, radial band (marked with a red arrow in rightmost column of **Figure 5**) is clearly shown in the GOME-2 NO<sub>2</sub> column, MODIS AOD, and CMAQ surface  $PM_{10}$  concentrations. Regional atmospheric circulation patterns forming this narrow band of pollutants can be more clearly seen in the hourly model output; an animation of this is available in the supplementary materials.
- Figure 6 shows another example of Type 2 narrow-band pollutants transport pattern from near Shanghai to Japan (marked with a red arrow), observed on April 5, 2014. The same band pattern can be seen from MODIS and VIIRS true color images in Figure 7. The white band and brownish plume offshore Shanghai might be a mixture of cloud and anthropogenic pollutants. This collocation may indicate the interaction of clouds and aerosols enhancement of cloud with aerosol as cloud condensation nuclei and/or formation of secondary PM associated with cloud in the region (Saide et al., 2015; B. Zhang et al.)
- al., 2015). Figure 6 exhibits another pollutant band over the open ocean along the cold front of the developed cyclone around northern Hokkaido (see Figure 7 for the location of the cold front). This band is clearly observed from MODIS AOD and CMAQ surface  $PM_{10}$  concentrations. Indeed, these two bands well represent an example of primary and secondary cold front passages (Bjerknes and Solberg, 1922). Although not shown, multi-year inspection of satellite observations and model

simulations revealed that such a narrow band of pollutants is quite common in East Asia. In fact, a narrow-band plume over the open ocean also appears in **Figure 4**. The occurrences of Type 2 pollutant transport pattern are further shown in **Figure 9** and discussed in Section 4.4.

# 4.3 Type 3: Stagnant high-pressure system

- When a high-pressure system is stationary or moving slowly over southern China or Yellow Sea, there is a high chance of a multi-day high  $PM_{10}$  concentration episode (Fig. 2c). We have classified such cases as Type 3. Figure 8 shows sequential plots of synoptic weather patterns and MODIS AOD from February 19–27, 2014, one of the worst high  $PM_{10}$  concentration episodes in China and Korea. An anticyclonic system centered around the South China Sea (marked with a red arrow) became stagnant for almost a week. On February 19, the MODIS AOD was already very high over northern China, as this
- region had already experienced a high PM<sub>10</sub> concentration episode in the previous week, while AOD over Korea was relatively low. On February 22, the high AOD extended to the Yellow Sea and the west coast of Korea, remaining high until February 26 when the stagnant high-pressure system was pushed out by the newly developing low-pressure system over Manchuria. On February 27, the low-pressure system over Manchuria prevailed, and the high-pressure system over the Yellow Sea moved to the western Pacific. Then, the AOD and surface PM concentrations over China and Korea began to
- decline.

Several previous studies have examined the pollutant episodes that occur under this type of stagnant condition, with a slowly-moving surface anticyclone. For instance, by investigating a high  $PM_{10}$  concentration episode in October 2008, Lee et al. (2013a) reported that the stagnant high-pressure system over Korea may play a decisive role in the accumulation of air pollutant in Seoul, Korea. Oh et al., (2015) also showed a strong positive geopotential height anomaly over Korea and Japan

- during multi-day high  $PM_{10}$  concentration episodes during 2001-2013. Ji et al. (2014) analyzed PM composition measurements from 20 monitoring sites in northern China and documented a high sulfur and nitrogen oxidation ratio, suggesting additional production of sulfates and nitrates with high emissions, under stagnant weather conditions. They proposed that the formation of secondary PM is one important mechanism in the formation of heavy air pollution episodes. Zhang et al. (2014) also reported that inorganic species and secondary organic aerosol components contribute to high air
- pollution episodes, most likely due to aqueous-phase processing under stagnant conditions with warm and humid air mass. We also note that the dissipation of high  $PM_{10}$  concentration events under Type 3 pattern is mechanically similar to the second step of Type 2 transport pattern. The development of a low-pressure system over Manchuria and the passage of a cold front function as a very efficient mechanism to remove high levels of air pollutants from China and Korea. The role of the Manchurian low-pressure system as a sweeping mechanism of pollutants warrants further investigation.

#### 30 4.4 PM<sub>10</sub> time series in the Seoul Metropolitan Area (SMA), Korea

**Figure 9** shows timeseries of  $PM_{10}$  concentrations and meteorological conditions in the SMA during the two cool seasons, November 2012 to May 2013 and November 2013 to May 2014. In each upper panel, the shaded areas represent minimum

and maximum  $PM_{10}$  concentrations from 105 surface monitoring sites in the SMA, with black dots representing all-site averages. Changes in surface pressure and wind speed are shown as red and blue dots, respectively, in the lower panels. The red asterisk (\*) symbol indicates days of Asian Dust guidance reported by the **KMA** as (http://www.kma.go.kr/weather/asiandust/observday.jsp).

- For each high PM<sub>10</sub> concentration episode, related pollutant transport pattern type, according to our classification (e.g., Types 1–3), is identified (Figure 9). This classification is subjectively conducted by examining individual sub-daily surface weather maps and satellite observations. While each type does not always occur alone, Figure 9 provides a good evidence to support the hypothesis that SMA PM concentrations are highly associated with meteorological conditions in East Asia. Note that transport pattern is not designated if there is no prevailing weather pattern. Below, several key observations from Figure
- 9 are briefly described.

In general, we notice a prominent correlation between sea level pressure and wind speed changes. In many cases, when sea level pressure rapidly decreases, we observe increased wind speed, which seems to be a typical signal of the passage of frontal zone associated with traveling low-pressure systems (i.e., systems initiated from Altai-Sayan cyclogenesis region). Such signals happen frequently across the entire cool season; see November 7, 11, 18, 25, and 27 and December 9, 11, 20,

and 27, 2013, as examples.

The signal of frontal passage is a good indicator of Type 2 and Type 3 pollutant transport patterns. Usually, PM concentrations are accumulated under high-pressure systems. Later, the frontal passage (i.e., sea level pressure drops and wind speed increases) tends to dissipate the Type 2 high PM concentration events. When this cleaning mechanism is weak or does not happen for multiple days (i.e., when the frontal passage associated with migratory low-pressure systems does not

sweep out the PM concentrations), we could observe a Type 3 pattern with a stagnant high-pressure system. Note that Type 1 pattern differs from Type 2 or Type 3, in terms of an early increase of wind speed during high  $PM_{10}$  concentration events. In many cases, the Type 1 pattern pushes pollutants from northern China to Korea and Japan. This pattern is also associated with the long-range transport of Asian Dust.

The synoptic weather pattern, associated with pollutant transport, is often very complicated, making it very challenging to

25 clearly separate one from another. In many cases, high PM concentration events in the SMA are caused by combinations of multiple synoptic weather patterns. However, one general observation can be summarized as follows. For persistent high PM concentration events, the existence of a prevailing high-pressure system is essential. On the other hand, a low-pressure system, which accompanies strong cold-frontal activity, functions not only as a strong transporting force, especially in the case of Asian Dust, but also as a cleaning mechanism for Type 2 and Type 3 patterns.

### 30 5 Summary and Conclusions

The present study examines pollutant transport patterns in East Asia and attempts to visualize them using all available satellite images and chemistry transport model output. It is shown that the spatial distribution of satellite-observed AOD,