# Peer review of "Synoptic perspectives on pollutant transport patterns observed by satellites over East Asia: Case studies with a conceptual model"

_Atmospheric Chemistry and Physics, 2016_

## Referee Comment (RC1) · Anonymous Referee #2 · 7 Oct 2016

General comments: As the authors have already acknowledged in the summary and conclusions section, this manuscript lacks of (if not none) quantitative analyses and provides no conclusive findings. There are a lot of jumps between reasoning steps in statements throughout the manuscript with no solid supporting evidence present. In fact, most part of the analyses are descriptive and subjective. There are many validated scientific tools out there but none of them are used here. Further, the entire manuscript is just a display of all kinds of "observations" assembled, with no scientific validation of the datasets used and no uncertainty checks on them. It is not convincing to the reviewer that the manuscript in its current shape can be an independent scientific paper. However, the datasets and the plots the manuscript collected/produced can be

useful supporting material for advanced quantitative analyses.

Specific comments: (1) What pollutants does this manuscript study for transport patterns? Satellite images/retrievals can be useful to track certain air pollution phenomena such as sand dust storms and forest fires. The fundamental reason for that is that the origin/source of sand dust and forest fire are obvious plus there is less secondary formation involved (Duncan et al. AE 2014). However it is not appropriate that the authors borrow the same method to analyze the transport of complex anthropogenic emissions and their contributions to regional surface PM concentration, especially if secondary formation are involved. It is difficult to image that satellite images/retrievals can separate the local and transport contributions to surface PM concentration at a certain receptor location. Without such a separation, how can a transport pattern be determined? (2) What uncertainties are with the satellite data used here? Air pollution is indexed by surface concentration of air pollutants. None of the satellite image or retrievals are direct observation of surface concentration of air pollutants. Satellite retrievals contain large uncertainties due to all kinds of factors such as cloud, aerosol layer, albedo, terrain etc. The uncertainty of AOD and $NO_2$ column data varies at different time and location. $NO_2$ retrievals are also dependent on what vertical profile used. There is no uncertainty check (or at least there is no QA/QC reported in the manuscript) on the satellite data used here. For example, in Figures 3&4, MODIS AOD has large areas with data missing, but at the same time and location, GOME-2 $NO_2$ has complete valid data available at those areas. Why this happened considering $NO_2$ are more sensitive to cloud and aerosol layers? (3) What uncertainties are with the modeling results that used to produce the simulated concentration maps? There is no evaluation reported for the model produced surface concentration maps. There are PM10 measurements shown in Figure 9, why don't the authors use these information to evaluate the simulations? Can the authors find more measurements such as at other locations to evaluate the simulations? The 2006 and 2007 emissions inventory are used for all the 2013-2015 simulations. How accurate the old inventory can be used to represent the more recent years without projection? Especially there was a

2008 worldwide economic recession/decline? And plus there is an air pollution control campaign carried out in China since 2013. (4) In addition, what ratios of PM2.5/PM10 are observed there? This is important information for meaningful analysis. (5) Satellite images are once (or twice)-per-day snap shots. Low-pressure system or frontal passage are moving fast, it can clean up all the things within just several hours. The arrival timing of such transport carried by these systems matters. What are the exact timing of the each panel in Figures 3-8? It would be very difficult to derive useful information from daily average maps for a rapid changing process. Satellite images can be taken before or after the arrival of cold front system at certain locations (it would be very rare the images were taken exactly at the arrival time). If it was taken before, the certain locations would be still under the control of the high-pressure system and would be more impacted by the local emissions. (6) Lifetime of NOx emissions is short. NO2 column data are usually used to represent NOx emissions (i.e. local emissions, not transported precursors). The association of NO2 column and AOD data are in fact showing the local contribution of anthropogenic emissions to PM concentrations instead of that being transported downstream by fast moving low-pressure system. (7) What mechanism can bring the lifted transported pollutants down to the surface with a fast moving low-pressure system? Are there different mechanisms working for different pollutants: dust, PM10, PM2.5 and other precursor pollutants? Column data doesn't differentiate surface density from the density above. Howe do the authors relate lifting transport to surface concentrations? (8) Transport patterns need to separate local contributions and those being transported, if unable to do so, how satellite images can help to identify any transport patterns? Models with appropriate tools can help, but in that case, first, modeling results of air pollutant concentrations should be extensively evaluated, inventory should be updated and modeled meteorology should be evaluated too.

---

## Referee Comment (RC2) · Anonymous Referee #1 · 7 Oct 2016

This paper presents an interesting analysis of the relationship between air pollution transport and synoptic weather in East Asia based on weather maps, PM measurements and regional chemical transport modelling. The overall topic of this paper fits to the scope of this journal. However, I think that in the current version the weather typing method was not so objective, and the numerical simulation needs to be improved and validated. Moreover, the discussion part was mainly descriptions on pollution episodes and its associated synoptic weather condition based on horizontal patterns. More quantitative results and in-depth analysis are needed to improve the scientific significance of this work. The paper would benefit from improvements along several main lines, which should amount to major revision.

[Figure]

Specific comments:

Conceptual classification of pollutant transport patterns in the work is too subjective. Numerical classification or other objective method is more applicable, especially in terms of pollution forecast.

As presented in Section 2, surface weather maps was utilized to determine regional pollutant transport patterns in the present work. However, long-range transport might be more related to 850-hPa and 700-hPa maps, especially for Asian Dust. Please justify why this work used surface maps.

WRF-CMAQ modelling: As described in the Section 3.3, the model was driven by NCEP GFS forecasts data. Here, reanalysis data like FNL with observation assimilation will improve the model performance in meteorology reproduction, and also transport of air pollutants. The authors mentioned that the model performance was presented in Kim et al. (2016). However, the modelling time period, meteorological input data and emission inventories used in the present work were all different from those in Kim et al. (2016). Thus, the model evaluation for this simulation (both meteorological parameters and pollutant concentrations) should be conducted and discussed in the manuscript. Observations at 105 surface monitoring sites (Section 4.4) can be used to validate the modelling results.

In Section 4, the authors only gave general pictures of horizontal distributions of surface pressure and satellite retrievals using combined plots. It makes no sense to repeatedly show the combined plots throughout the article (Figure 3-8). For the same reason, it's unnecessary to include too much introduction on the geo-referencing method in the manuscript. This section can be substantially improved by more in-depth analysis based on modelling results. Discussion on vertical structure/atmospheric stratification and its impacts on pollutant transport and dispersion will make more sense.

Warm conveyor belts (WCBs), which are associated with cyclones, are crucial in the long-range transport of air pollutants over East Asia. However, in this work, there is

no discussion on WCBs. To clarify the relation between synoptic weather and pollutant transport, more focus should to be paid on WCBs.

Most part of Section 4 is descriptive, lacking of quantitative analysis and constructive conclusions. The study period is less than two years, leading to the fact that the discussions are more like descriptions on pollution episodes, which has been addressed in detail by many existing studies. In addition, many conclusions need more support, for instance, the authors concluded that the PM pollution on December 30–31 was caused by anthropogenic emissions and Asian Dust transport without enough evidences. Checking the dust emission rate in the model and observed PM2.5/PM10 ratio can provide more useful information and supportive evidences. Besides, it should be noted that some pollution cases cannot be totally attributed to the transport. Some kinds of synoptic weather pattern might give rise to specific meteorological conditions (strong radiation, high air temperature, less precipitation and stable boundary layer) that favor the formation of secondary pollutions (NO2 and PM) or the accumulation of locally-emitted pollutants. The aforementioned uncertainties should be discussed and clarified here. It is actually insignificant to describe the individual pollution episode and its transport pathway in this work. Long-term dataset and statistical analysis could further strengthen the quantitative conclusions and improve the scientific importance.

Technical Corrections: Page 5, Line 23: NIER and KMA need to be specified when they are presented for the first time.

Page 6, Line 4-6: surface PM10 concentrations and KMA weather map are introduced twice in Section 3.1 and 3.2. Section 3.1 is too short to be a section, and the descriptions on graphical technique are redundant.

The quality of figures was too bad. The contour labels and legends in figures are not clear enough.

---

## Author Comment (AC1) · 30 Jan 2017

**Authors' response to the review comments**

**"Synoptic perspectives on pollutant transport patterns observed by satellites over East Asia: Case studies with a conceptual model" by Kim et al.**

**General responses**

The authors express their appreciation to the two reviewers and the editor. We believe that their comments are very productive and substantially contributed to improving the manuscript. We offer general responses and point-by-point responses to the issues and comments addressed by reviewers. Reviewers' comments are shown in italics.

Here are responses for main comments.

(1)      Lack of quantitative analysis (Reviewer 1 & 2)

We do understand the reviewers' concern that current manuscript is lack of numerical quantification. It may be our fault because we tried to separate the conceptual approach and numerical approach in current manuscript and concurrent/following studies. In strong agreement with reviewer's comments, we have included more complete model performance evaluations to make this manuscript a standalone research, but also tried to stay in the original scope of the study.

Here are the list of major changes in the manuscript

1.  Main model simulations are replaced to the NOAA FNL meteorology-initiated simulations. Both GFS-initiated and FNL-initiated simulations are analyzed.
2.  Multiple emission inventories are tested in response to the reviewer's comment on the feasibility of old emission inventories. Two sets of combinations for international and South Korean emission inventories are utilized.
3.  We expanded model performance evaluation with further detailed analyses. The additional analyses confirm that model simulations showed very good performance compared to surface observations in their spatial and temporal variation.

In addition, we like to clarify the scope of this study. This study hypothesizes a conceptual model to understand the correlation between regional pollution and synoptic meteorology. We designed this study to suggest control mechanisms of pollution's development and dissipation associated with synoptic systems (and visible evidences from satellites), so it can be further used to design climatological classification over longer term. However, we did not include numerical classification method itself in the current version of the study. Numerical classification, using the conceptual model suggested in this study, is an ongoing next step. As many researchers may agree, the issue of regional air quality in East Asia is very complicated, and we believe that proper classification should be based on conceptual understanding of underlying physics and chemistry.

We do not think that one study can solve all the questions of recent air quality issues in East Asia, especially on the puzzle why East Asia has experienced severe haze events during cold seasons in recent years even though there has been considerable reduction of anthropogenic emissions released from China. This study focuses on the role of synoptic weather among many necessary studies. Hopefully, this study provides a few missing links towards understanding regional air quality and meteorology.

(2)     Use of satellite observation (Reviewer 2)

We do not agree that satellites are useless in monitoring regional air quality in East Asia. We understand and agree with the reviewer's concern on the limitation of satellite monitoring, but we also believe in the strong advantages of using satellite data for regional air quality monitoring. Over decades, numerous satellite products have helped monitoring regional pollutants and their precursors.

Satellite monitoring can be very useful because *it can evaluate model outputs and can constrain model inputs*. Satellite product is limited when it was used alone. However, satellite products can provide a good synergy when combined and interpreted with additional information. This study demonstrates the capability of an integrated system of satellite, model and weather analysis to advance understanding in regional air quality. We would not reject the use of satellite data due to its uncertainty.

(3)     Scientific importance (Reviewer 2)

In recent years, frequent occurrence of severe haze events in East Asia is one of the most serious public concerns in this region. The reason for the increased haze events is still unknown and very puzzling. Since recent space-borne monitoring of Chinese anthropogenic emissions indicated a decreasing trends in $NO_x$ and $SO_2$ emissions (Duncan et al., 2016), the role of meteorology has getting more attention, as addressed in the manuscript and the studies mentioned by the reviewer #1. Our manuscript focuses on the possible role of meteorology, especially by the routine passages of synoptic systems, on the formation and removal of regional pollutions in East Asia. This manuscript addresses the importance and governing characteristics of meteorology in the air quality of East Asia.

Duncan et al., 2016: A space-based, high-resolution view of notable changes in urban NOx pollution around the world, doi:10.1002/2015JD024121

We also provide specific responses below.

**Anonymous Referee #2**

*General comments: As the authors have already acknowledged in the summary and conclusions section, this manuscript lacks of (if not none) quantitative analyses and provides no conclusive findings. There are a lot of jumps between reasoning steps in statements throughout the manuscript with no solid supporting evidence present. In fact, most part of the analyses are descriptive and subjective. There are many validated scientific tools out there but none of them are used here. Further, the entire manuscript is just a display of all kinds of "observations" assembled, with no scientific validation of the datasets used and no uncertainty checks on them. It is not convincing to the reviewer that the manuscript in its current shape can be an independent scientific paper.*

Thanks for the comment. We apologize that we did not provide more specific the model evaluation previously. We provide more detailed model evaluation in the new manuscript. For example, Figure R1 & R2 show comparison of surface $PM_{10}$, ozone and $NO_2$ concentrations during November 2013. Monthly comparisons during all study period are included in the supplementary material.

We do not agree that this study provides no conclusive findings. In fact, our study addresses very important implication to understand the discrepancies between decreased anthropogenic emissions and increased severe haze events in the East Asian countries. As described in the conclusion, synoptic weather systems in this region are likely one of the main mechanisms to control the accumulation (with high pressure systems) and dissipation (by cyclones) of pollutants. Actually, this finding is connected to our next study (Kim et al., 2017c) that demonstrates that recent increase of surface particulate matter concentration in South Korea is fully explained by the interannual variation of surface wind speed, so the strength of midlatitude ventilation plays a crucial role in regional air quality in East Asia (See Figure R3). This implies that the role of synoptic weather system in East Asia is as large as the rapid change of anthropogenic emissions. We believe current study provides very important link to connect this region's air quality to changes of meteorology and/or climate.

*However, the datasets and the plots the manuscript collected/produced can be useful supporting material for advanced quantitative analyses.*

Thanks for the comment. We hope this study helps other quantitative studies by linking short-term modeling and observations to long-term climate statistics. We do respect values of quantitative statistics of long-term analysis, but also believe that the short-term and in situ observations should not be ignored. For example, the shallow band-shaped plumes, described in the manuscript, were rarely identified in the monthly time scale. In daily satellite observations, they looked like plumes coming out of Shanghai region. More detailed information, using hourly model and weather chart analysis, it was revealed that they were possibly from northern China. It was an interesting case because it might imply that traditional trajectory classifications based on coarse temporal resolution data needed careful interpretation on their source-receptor relation.

*Specific comments:*

*(1) What pollutants does this manuscript study for transport patterns? Satellite images/retrievals can be useful to track certain air pollution phenomena such as sand dust storms and forest fires. The fundamental reason for that is that the origin/source of sand dust and forest fire are obvious plus there is less secondary formation involved (Duncan et al. AE 2014). However it is not appropriate that the authors borrow the same method to analyze the transport of complex anthropogenic emissions and their contributions to regional surface PM concentration, especially if secondary formation are involved.*

Thanks for the comment. Our main focus is the regional scale transport of anthropogenic particulate matters in East Asia. We focused on not only "transport" of pollutant plume but also analyzed how synoptic patterns in this region were associated with "development" and "dissipation" of pollutants.

We do not agree that satellite observations are ineffective to monitor anthropogenic pollutants in East Asia. Simply, strength of anthropogenic pollutants from China is considerably stronger than that by natural particulate matter emissions in North America. Satellite observations have been used to monitor anthropogenic pollutant plumes in East Asia for decades (Guo et al., 2011; Wang et al., 2011). Satellite retrieved $NO_2$ column densities also had been used to detect anthropogenic emissions in many previous studies (Richter et al., 2005). Satellite retrievals have limitation to separate detailed components of air pollutants, but when combined with model simulations, they have a good synergy to interpret chemical components in the atmosphere.

Guo et al., 2011: Spatio-temporal variation trends of satellite-based aerosol optical depth in China during 1980-2008, doi:10.1016/j.atmosenv.2011.03.068

Richter et al., 2005: Increase in tropospheric nitrogen dioxide over China observed from space, doi:10.1038/nature04092

Wang et al., 2011: Verification of anthropogenic emissions of China by satellite and ground observations, doi:10.1016/j.atmosenv.2011.08.054

*It is difficult to image that satellite images/retrievals can separate the local and transport contributions to surface PM concentration at a certain receptor location. Without such a separation, how can a transport pattern be determined?*

Thanks for the comment. The main advantage of satellite data is to provide wide coverage. For example, if satellite data show heavy aerosols over the Yellow Sea wherein westerly wind prevails repeatedly, it clearly means that those aerosols are consistently transported from continental sources or formed from the thus transported precursors. Separation of upwind- and downwind- locations of emission sources is commonly used to understand the attribution of pollution source.

*(2) What uncertainties are with the satellite data used here? Air pollution is indexed by surface concentration of air pollutants. None of the satellite image or retrievals are direct observation of surface concentration of air pollutants. Satellite retrievals contain large uncertainties due to all kinds of factors such as cloud, aerosol layer, albedo, terrain etc. The uncertainty of AOD and NO2 column data varies at different time and location. NO2 retrievals are also dependent on what vertical profile used. There is no uncertainty check (or at least there is no QA/QC reported in the manuscript) on the satellite data used here. For example, in Figures 3&4, MODIS AOD has large areas with data missing, but at the same time and location, GOME-2 NO2 has complete valid data available at those areas. Why this happened considering NO2 are more sensitive to cloud and aerosol layers?*

Thanks for the comment. Currently, there are two operational GOME-2 instruments (onboard MetOp-A & MetOp-B satellite) with slightly different coverage, so they are designed to fill each other's coverage gap. Combined coverage from two GOME-2 instruments shows better spatial coverage compared to MODIS. Please, refer to Figure R3 for the daily coverage of GOME-2. For comparison of these satellite images, model simulations are displayed for 12PM local time in figures, but spatial- and temporal-collocation is accurately considered for any numerical evaluation.

MODIS has been operational more than 15 years. MODIS AOD has been intensively evaluated, and believed to be one of most reliable products. We know all satellite products have their own uncertainty, but we do not think MODIS, OMI and GOME-2 have any quality issue susceptible to total rejection. Uncertainty of each product is addressed in the reference paper, and we added more descriptions of each products' data quality in the manuscript.

*(3) What uncertainties are with the modeling results that used to produce the simulated concentration maps? There is no evaluation reported for the model produced surface concentration maps. There are PM10 measurements shown in Figure 9, why don't the authors use these information to evaluate the simulations? Can the authors find more measurements such as at other locations to evaluate the simulations?*

Thanks for the comment. We apologize that we did not include model evaluation results. We have included 19 months evaluation in fine temporal resolution. Please, refer to Figure R1 & R2, or supplementary material.

*The 2006 and 2007 emissions inventory are used for all the 2013-2015 simulations. How accurate the old inventory can be used to represent the more recent years without projection? Especially there was a 2008 worldwide economic recession/decline? And plus there is an air pollution control campaign carried out in China since 2013.*

Thanks for the comment. We have included additional simulations using two sets of emission inventories for Asia and domestic South Korea: (1) INTEX-B 2006 (For Asia) and CAPSS 2007 (South

Korea0, and (2) MICS-Asia 2010 (Asia) and CAPSS 2010 (South Korea). In current study, we illustrate the spatial distribution and transport of pollutants in relation to synoptic meteorology - varying strength of emissions did not result in significant differences. In addition, ironically, $NO_x$ emissions from China have shown dramatic changes, and $NO_x$ emission level during the study period was similar to that of early years. Please, refer to Figure R5 for the chage of $NO_2$ column densities from multiple satelites over East Asia. Model evaluation also shows no emission inventory issues.

*(4) In addition, what ratios of PM2.5/PM10 are observed there? This is important information for meaningful analysis.*

Thanks for the comment. We concur that the $PM_{2.5}$ to $PM_{10}$ mass ratio is a strong indicator for pollutant components. We expanded the discussion on Dec. 31, 2013 case and included analysis on the change of PM2.5/PM10 ratios during the event in the supplementary material.

*(5) Satellite images are once (or twice)-per-day snap shots. Low-pressure system or frontal passage are moving fast, it can clean up all the things within just several hours. The arrival timing of such transport carried by these systems matters. What are the exact timing of the each panel in Figures 3-8? It would be very difficult to derive useful information from daily average maps for a rapid changing process. Satellite images can be taken before or after the arrival of cold front system at certain locations (it would be very rare the images were taken exactly at the arrival time). If it was taken before, the certain locations would be still under the control of the high-pressure system and would be more impacted by the local emissions.*

All instruments used in this study are onboard polar orbital satellites, so they have specific local pass time. Aqua (1:30PM) and Terra (10:30AM) for MODIS, Aura (1:30PM) for OMI, and MetOp-A & MetOp-B (9:30AM) for GOME-2) as mentioned in the manuscript.

Satellites provide snap shots, so we may not determine where the pollutant plume comes from only with satellite image. However, it is very common to use additional information, such as model or weather map, to read flow patterns, then determine its transport pathways. For example, one can establish conceptual relationship between repeated model simulation with pollutant plumes over the Yellow Sea, and satellite's corresponding image of pollution. Both these information sources corroborated that wind direction was westerly or northwesterly. We feel confident to conjecture that pollutants in this case came from western sources since there is no dominant $NO_x$ emission sources over the ocean.

*(6) Lifetime of NOx emissions is short. NO2 column data are usually used to represent NOx emissions (i.e. local emissions, not transported precursors). The association of NO2 column and AOD data are in fact showing the local contribution of anthropogenic emissions to PM concentrations instead of that being transported downstream by fast moving low-pressure system.*

Thanks for the comment. For the $NO_2$ plume extended from mainland China (Figure 6 in the manuscript), there is no significant $NO_x$ emission sources over the Yellow Sea except emissions from traveling ships. Considering prominent northwesterly wind direction, it is fairly reasonable that this plume is transported and extended from sources in China.

*(7) What mechanism can bring the lifted transported pollutants down to the surface with a fast moving low-pressure system? Are there different mechanisms working for different pollutants: dust, PM10, PM2.5 and other precursor pollutants? Column data doesn't differentiate surface density from the density above. Howe do the authors relate lifting transport to surface concentrations?*

Thanks for the comment. Lifting mechanism associated with cold frontal passage does not mean simple upward movement of a solid object. It is associated with complicated upward motion in warm side and strong downward motion in cold side. It is also associated lateral detrainment at many levels. Please,

refer to Figure R5 (or meteorology text book) for the structure of cold frontal passages and air flows. We cannot simply argue that all the pollutants are lifted up by cold front and have nothing with surface concentration.

There is no conclusive studies to explain how high these anthropogenic plumes travel over the Yellow Sea. This may be different from the case of Asian Dust which cannot travel long-range unless it is fully aloft. At least, model simulation shows enhanced particulate matter plumes extending from the surface up to (and above) boundary layer, being pushed by the cold front. In fact, the uplifting capability of cold front is an interesting topic, but it is out of current study's scope since we do not have in-situ observations in current study period. We will pursue it later with campaign data over the Yellow Sea – e.g. KORUS-AQ.

*(8) Transport patterns need to separate local contributions and those being transported, if unable to do so, how satellite images can help to identify any transport patterns? Models with appropriate tools can help, but in that case, first, modeling results of air pollutant concentrations should be extensively evaluated, inventory should be updated and modeled meteorology should be evaluated too.*

Thanks for the comment. This comment is exactly what we like to demonstrate in this study. Most of satellite observations are short term snap shots, and do not provide much information when used alone. However, when remote-sensed information is combined with chemistry transport model and synoptic weather analysis, they have very strong synergy and can provide better understanding. We apologize for the lack of model evaluation for the initial manuscript. As reviewer commented we have included detailed model performance evaluations to confirm the good performance of current model used in this study.

**Conclusive remarks**

We again appreciate reviewers' comments. We think that most of reviewers' comments are helpful and relevant. We tried our best to cover most of the points addressed by reviewers, and, at the same time, we tried to keep the scope of this study clear and concise. Additional analysis are or will be available in following studies.

1. Quantitative estimation of contributions from regional pollutants and precursors (Kim et al., 2017a)
2. Formation of secondary aerosol (Kim et al., 2017b)
3. Long-term correlation of regional weather and pollution (Kim et al., 2017c)

Kim, E., C. Bae, **H. C. Kim**, J. H. Cho, B.-U. Kim, and S. Kim, 2017a: Regional Contributions to Particulate Matter Concentration in the Seoul Metropolitan Area, Korea: Seasonal Variation and Sensitivity to Meteorology and Emissions Inventory, *Atmospheric Chemistry and Physics Discussion,* doi:10.5194/acp-2016-1114

Kim, B.-U., C. Bae, **H. C. Kim**, E. Kim, and S. Kim, 2017b: Spatially and Chemically Resolved Source Apportionment Analysis: Case Study of High Particulate Matter Event in the Seoul Metropolitan Area, South Korea, during late February 2014, *under review*

**Kim, H. C.**, S. Kim, B.-U. Kim, C.-S. Jin, R. Park, C. Bae, M. Bae, and A. Stein, 2017c: Recent increase of surface particulate matter concentrations in the Seoul Metropolitan Area, Korea, *under reivew*

[Figure]

**Figure R1 Time series of surface PM$_{10}$, O$_3$ and NO$_2$ concentrations from model and surface monitoring sites during November 2013 over the Seoul, Korea. Observations from 107 sites are compared. Simulation is conducted using the INTEX-B 2006 emission inventory over Asia and CAPSS 2007 over South Korea. Additional comparisons are provided in the supplemental materials.**

[Figure]

**Figure R2 Same with Figure R1 except emission inventories from MICS-2010 and CAPSS 2010.**

[Figure]

**Figure R3. Normalized anomalies of annual mean surface PM concentration (a), and annual mean 10-m wind speed (b). Red (blue), pink (light blue), and dashed pink (light blue) lines indicate anomalies of modeled PM concentrations averaged over the 9-km domain-wide, Korea (land pixels), and the SMA regions for PM concentrations (for wind speed). Circles indicate observations of surface PM concentrations (257 sites over South Korea) and wind speed (79 sites over South Korea). The scatter plot (c) shows a least square regression fit between normalized anomalies of surface PM concentrations and wind speed from the model (9-km domain average). (Kim et al., 2017c)**

[Figure]

**Figure R4 Spatial coverage of GOME-2 onboard MetOp-A (upper) and MetOp-B (lower).**

[Figure]

**Figure R5 Time series of GOME, SCIAMACHY, OMI, and GOME-2 NO₂ VCD for (a) China, (b) the BTH region, (c) Shanghai, (d) Hong Kong, (e) Korea, and (f) Japan. NO₂ VCD is shown in the upper panels, and 12-month moving averages, normalized to the 2010 mean, are shown in the lower panels. (Kim et al. 2016, under review)**

[Figure]

**Figure R6** https://www.britannica.com/science/thunderstorm/images-videos/Evolution-of-a-gust-front-During-a-thunderstorm-a-large/19393

---

## Author Comment (AC2) · 31 Jan 2017

**Authors' response to the review comments**

**"Synoptic perspectives on pollutant transport patterns observed by satellites over East Asia: Case studies with a conceptual model" by Kim et al.**

**General responses**

The authors express their appreciation to the two reviewers and the editor. We believe that their comments are very productive and substantially contributed to improving the manuscript. We offer general responses and point-by-point responses to the issues and comments addressed by reviewers. Reviewers' comments are shown in italics.

Here are responses for main comments.

(1) Lack of quantitative analysis (Reviewer 1 & 2)

We do understand the reviewers' concern that current manuscript is lack of numerical quantification. It may be our fault because we tried to separate the conceptual approach and numerical approach in current manuscript and concurrent/following studies. In strong agreement with reviewer's comments, we have included more complete model performance evaluations to make this manuscript a standalone research, but also tried to stay in the original scope of the study.

Here are the list of major changes in the manuscript

- 1. Main model simulations are replaced to the NOAA FNL meteorology-initiated simulations. Both GFS-initiated and FNL-initiated simulations are analyzed.
- 2. Multiple emission inventories are tested in response to the reviewer's comment on the feasibility of old emission inventories. Two sets of combinations for international and South Korean emission inventories are utilized.
- 3. We expanded model performance evaluation with further detailed analyses. The additional analyses confirm that model simulations showed very good performance compared to surface observations in their spatial and temporal variation.

In addition, we like to clarify the scope of this study. This study hypothesizes a conceptual model to understand the correlation between regional pollution and synoptic meteorology. We designed this study to suggest control mechanisms of pollution's development and dissipation associated with synoptic systems (and visible evidences from satellites), so it can be further used to design climatological classification over longer term. However, we did not include numerical classification method itself in the current version of the study. Numerical classification, using the conceptual model suggested in this study, is an ongoing next step. As many researchers may agree, the issue of regional air quality in East Asia is very complicated, and we believe that proper classification should be based on conceptual understanding of underlying physics and chemistry.

We do not think that one study can solve all the questions of recent air quality issues in East Asia, especially on the puzzle why East Asia has experienced severe haze events during cold seasons in recent

years even though there has been considerable reduction of anthropogenic emissions released from China. This study focuses on the role of synoptic weather among many necessary studies. Hopefully, this study provides a few missing links towards understanding regional air quality and meteorology.

**(2) Use of satellite observation (Reviewer 2)**

We do not agree that satellites are useless in monitoring regional air quality in East Asia. We understand and agree with the reviewer's concern on the limitation of satellite monitoring, but we also believe in the strong advantages of using satellite data for regional air quality monitoring. Over decades, numerous satellite products have helped monitoring regional pollutants and their precursors.

Satellite monitoring can be very useful because *it can evaluate model outputs and can constrain model inputs*. Satellite product is limited when it was used alone. However, satellite products can provide a good synergy when combined and interpreted with additional information. This study demonstrates the capability of an integrated system of satellite, model and weather analysis to advance understanding in regional air quality. We would not reject the use of satellite data due to its uncertainty.

**(3) Scientific importance (Reviewer 2)**

In recent years, frequent occurrence of severe haze events in East Asia is one of the most serious public concerns in this region. The reason for the increased haze events is still unknown and very puzzling. Since recent space-borne monitoring of Chinese anthropogenic emissions indicated a decreasing trends in NOx and SO2 emissions (Duncan et al., 2016), the role of meteorology has getting more attention, as addressed in the manuscript and the studies mentioned by the reviewer #1. Our manuscript focuses on the possible role of meteorology, especially by the routine passages of synoptic systems, on the formation and removal of regional pollutions in East Asia. This manuscript addresses the importance and governing characteristics of meteorology in the air quality of East Asia.

Duncan et al., 2016: A space-based, high-resolution view of notable changes in urban NOx pollution around the world, doi:10.1002/2015JD024121

We also provide specific responses below.

**Anonymous Referee #1**

This paper presents an interesting analysis of the relationship between air pollution transport and synoptic weather in East Asia based on weather maps, PM measurements and regional chemical transport modelling. The overall topic of this paper fits to the scope of this journal. However, I think that in the current version the weather typing method was not so objective, and the numerical simulation needs to be improved and validated. Moreover, the discussion part was mainly descriptions on pollution episodes and its associated synoptic weather condition based on horizontal patterns. More quantitative results and in-depth analysis are needed to improve the scientific significance of this work. The paper would benefit from improvements along several main lines, which should amount to major revision.

Thanks for the comment. We apologize that we did not provide more specific the model evaluation previously. We provide more detailed model evaluation in the new manuscript. For example, Figure R1 &R2 show comparison of surface PM10, ozone and NO2 concentrations during November 2013. Monthly comparisons during all study period (19 months) are included in the supplementary material.

**Specific comments:**

**Conceptual classification of pollutant transport patterns in the work is too subjective. Numerical classification or other objective method is more applicable, especially in terms of pollution forecast.**

Thanks for the comment. As mentioned in the general response, this study is designed to provide an intermediate step to advance towards long-term numerical classification. We believe that numerical classification should be designed on the proper understanding of underlying physics and chemistry. In this study, we suggest to go back to the basic and to establish a conceptual model first between systematic occurrence of synoptic patterns and regional pollutant's accumulation and dissipation. Numerical classification, using the conceptual model suggested in this study, is an ongoing next step.

**As presented in Section 2, surface weather maps was utilized to determine regional pollutant transport patterns in the present work. However, long-range transport might be more related to 850-hPa and 700-hPa maps, especially for Asian Dust. Please justify why this work used surface maps.**

Thanks for the comment. While we agree that 850-mb weather chart is more suited for the analysis of long-range transport of pollutants, especially for the transport of Asian Dust, we have two reasons to provide surface charts in current study. First, unlike Asian dust cases, the transport pathways of anthropogenic pollutants from Chinese cities and industrial areas are not fully understood. Anthropogenic pollutants, especially secondary aerosols, have smaller size distribution compared to Asian dust, and they have higher chance to transport even in lower altitude. We could not exclude the importance of surface synoptic displacement. Second, more practically, most of 850-mb charts are available only for every 12 hours. They did not have enough temporal resolution to be combined with hourly model simulations to analyze detailed transport pathways. For further information, we have included available 850-hPa weather analysis charts in the supplementary materials.

**WRF-CMAQ modelling: As described in the Section 3.3, the model was driven by NCEP GFS forecasts data. Here, reanalysis data like FNL with observation assimilation will improve the model performance in meteorology reproduction, and also transport of air pollutants.**

We agree that observation-assimilated FNL might provide better meteorology for pollution transport study. Actually, it was a part of our original plan, and we have completed additional simulations using the FNL meteorology during the review process. Results are provided with model performance evaluations. We confirm, however, there is no significant change in our conclusion by the choice of meteorology initiation.

The authors mentioned that the model performance was presented in Kim et al. (2016). However, the modelling time period, meteorological input data and emission inventories used in the present work were all different from those in Kim et al. (2016). Thus, the model evaluation for this simulation (both meteorological parameters and pollutant concentrations) should be conducted and discussed in the manuscript. Observations at 105 surface monitoring sites (Section 4.4) can be used to validate the modelling results.

Kim et al. (2016) was referenced to show the model's physical configurations and general performance. We have included additional model performance evaluations. We apologize for the confusion.

In Section 4, the authors only gave general pictures of horizontal distributions of surface pressure and satellite retrievals using combined plots. It makes no sense to repeatedly show the combined plots throughout the article (Figure 3-8). For the same reason, it's unnecessary to include too much introduction on the geo-referencing method in the manuscript. This section can be substantially improved by more in-depth analysis based on modelling results. Discussion on vertical structure/atmospheric stratification and its impacts on pollutant transport and dispersion will make more sense.

Plots are presented to provide cases of different synoptic patterns corresponding to the classification. We recommend readers to also check animated plots included in the supplementary material since they provide more information on the pollutants' movement.

Geo-referencing technique was mentioned only two times in the main context. Although the technique itself has no importance in a scientific point of view, it is very practical for readers from operational institute. We have received several requests for technical details, including actual code, so they were included in the Appendix and supplementary material.

Satellite observations are excellent for spatial distribution but is technically limited to the evaluation of vertical structure unless they are designed for limb scanning. Figure R3 demonstrates sequences of modeled PM10 concentration vertical profiles during Dec. 5-7, 2013 episode, along a cross section over the Yellow Sea. They illustrate the propagation and the vertical extension of intensive pollutant plumes over the Yellow Sea pretty well. Its evaluation, however, is limited in current study. We believe vertical structures can be better analyzed using *in-situ* measurements.

We are also analyzing related vertical structure using *in-situ* aircraft measurements from the 2016 KORUS-AQ campaign. We cannot include any analysis here because they are out of the study period, and also we are not allowed to release any campaign observations before June 2017.

**Warm conveyor belts (WCBs), which are associated with cyclones, are crucial in the long-range transport of air pollutants over East Asia. However, in this work, there is no discussion on WCBs. To clarify the relation between synoptic weather and pollutant transport, more focus should to be paid on WCBs.**

Thanks for the comment. We included a discussion of the WCB. WCB describes the enhancement of pollutant concentration inside the warm sector – in front of cold front. Initially, we did not use the concept of WCB to avoid unnecessary confusion. While the WCB usually explains strong transport mechanism (both vertically and horizontally), it is quickly followed by strong removal of pollutants due to frontal activities. We have describes the dual role of low pressure systems – transport by WCB and effective removal of local pollutants.

**Most part of Section 4 is descriptive, lacking of quantitative analysis and constructive conclusions. The study period is less than two years, leading to the fact that the discussions are more like descriptions on pollution episodes, which has been addressed in detail by many existing studies.**

We agree that there have been previous studies to demonstrate pollution episodes, but they are mostly for a short campaign period (compared to 2 year periods in current study), and the use of satellite data is mostly limited. In this study, we introduce more integrated system to demonstrate surface observations, model and satellite observations combined with synoptic weather analysis altogether. For the use of satellite data, there are few studies to demonstrate NO2 column densities (e.g. as a proxy of anthropogenic emission plumes) from multiple satellites in daily basis. We believe this study provide a very unique platform for regional air quality analysis.

**In addition, many conclusions need more support, for instance, the authors concluded that the PM pollution on December 30–31 was caused by anthropogenic emissions and Asian Dust transport without enough evidences. Checking the dust emission rate in the model and observed PM2.5/PM10 ratio can provide more useful information and supportive evidences.**

Thanks for the comment. We concur that the  $PM_{2.5}$  to  $PM_{10}$  mass ratio is a strong indicator for pollutant components, especially for the Asian dust. We expanded the discussion on Dec. 31, 2013 case and included analysis on the change of  $PM_{2.5}/PM_{10}$  ratios during the event. Figure R4 shows changes of  $PM_{2.5}$ ,

PM10 and theirs ratios observed at the Bulkwang supersite and several Chinese surface sites during the Dec. 31 episode. The PM2.5 to PM10 ratios were high in the earlier development of the high PM episode, but became lower in the later stage, implying that this case is likely a mixture of anthropogenic pollution (earlier) and Asian dust (later), as described in the manuscript. Observations from Chinese sites also confirm the possibility of Asian dust case, showing lower PM2.5 to PM10 ratios in northern China. On the other hand, Figure R5 demonstrates a case of strong intrusion of Asian dust (Mar. 18, 2014), showing a high PM10 concentration with lower PM2.5 to PM10 ratios in Bulkwang supersite at Seoul, Korea. Chinese sites also indicate low PM2.5 to PM10 ratios showing likely Asian dust signal.

**Besides, it should be noted that some pollution cases cannot be totally attributed to the transport. Some kinds of synoptic weather pattern might give rise to specific meteorological conditions (strong radiation, high air temperature, less precipitation and stable boundary layer) that favor the formation of secondary pollutions (NO2 and PM) or the accumulation of locally-emitted pollutants. The aforementioned uncertainties should be discussed and clarified here.**

Thanks for the comment. We agree with the reviewer's comment that some kinds of synoptic weather pattern is associated with specific meteorological condition that is favorable to the formation of secondary pollution or the accumulation of local pollutants. It is important especially in the case of stagnant higher pressure system where all East Asian countries are under the similar meteorological condition. We already mentioned this point in the conclusion that it is not easy to separate regional transport from local development without a help of a proper chemistry transport modeling. We further clarified the point.

For this point, we have further quantitative analysis to separate the contributions from local and international emission sources to the surface concentration of particulate matter in South Korea (Kim et al., 2017a). Sensitivity to the choice of meteorological model and emission inventories are also discussed in the same study. Indeed, this is more complicated issue if we consider the formation of secondary aerosol during transport. Kim et al. (2017b) also discusses possibility of secondary aerosol formation by precursors from different regional emission sources. Finally, we like to clarify that this study does not have any conclusion on the attribution or responsibility of regional emissions sources to each Asian countries. We better like to understand the control mechanism.

**It is actually insignificant to describe the individual pollution episode and its transport pathway in this work. Long-term dataset and statistical analysis could further strengthen the quantitative conclusions and improve the scientific importance.**

While simply describing individual pollution episodes may not be important, we did not describe them without reason. We tried to demonstrate how well pollutants are accumulated under high pressure systems, and how effectively low pressure systems sweep out high concentrations. Actually, this finding is connected to our next study (Kim et al., 2017c) that demonstrates that recent increase of surface particulate matter concentration in South Korea is fully explained by the interannual variation of surface wind speed, so the strength of midlatitude ventilation plays a crucial role in regional air quality in East Asia (See Figure R6). This implies that the role of synoptic weather system in East Asia is as large as the rapid change of anthropogenic emissions. We believe current study provides very important link to connect this region's air quality to changes of meteorology and/or climate.

**Technical Corrections: Page 5, Line 23: NIER and KMA need to be specified when they are presented for the first time.**

**Corrected.**

Page 6, Line 4-6: surface PM10 concentrations and KMA weather map are introduced twice in Section 3.1 and 3.2.

**Corrected.**

**Section 3.1 is too short to be a section, and the descriptions on graphical technique are redundant.**

Section 3.1 is to separate observational and analytical data from satellite and model data. Description on the geo-referencing technique based included based on several requests of operational forecast agencies. If the reviewer still wants to remove it, we will move it to the supplementary material section.

**The quality of figures was too bad. The contour labels and legends in figures are not clear enough.**

As described in the Appendix, overlapping of weather chart was done pixel-by-pixel graphical transformation, so, technically, it cannot be enhanced beyond its original pixel resolution. Instead, we have included original weather chart graphics files for readers who like to read actual pressure contour labels.

**Conclusive remarks**

We again appreciate reviewers' comments. We think that most of reviewers' comments are helpful and relevant. We tried our best to cover most of the points addressed by reviewers, and, at the same time, we tried to keep the scope of this study clear and concise. Additional analysis are or will be available in following studies.

- 1. Quantitative estimation of contributions from regional pollutants and precursors (Kim et al., 2017a)
- 2. Formation of secondary aerosol (Kim et al., 2017b)
- 3. Long-term correlation of regional weather and pollution (Kim et al., 2017c)
- Kim, E., C. Bae, H. C. Kim, J. H. Cho, B.-U. Kim, and S. Kim, 2017a: Regional Contributions to Particulate Matter Concentration in the Seoul Metropolitan Area, Korea: Seasonal Variation and Sensitivity to Meteorology and Emissions Inventory, *Atmospheric Chemistry and Physics Discussion*, doi:10.5194/acp-2016-1114
- Kim, B.-U., C. Bae, H. C. Kim, E. Kim, and S. Kim, 2017b: Spatially and Chemically Resolved Source Apportionment Analysis: Case Study of High Particulate Matter Event in the Seoul Metropolitan Area, South Korea, during late February 2014, *under review*
- Kim, H. C., S. Kim, B.-U. Kim, C.-S. Jin, R. Park, C. Bae, M. Bae, and A. Stein, 2017c: Recent increase of surface particulate matter concentrations in the Seoul Metropolitan Area, Korea, *under reivew*